# Physical Activity Behaviour and Motivation During and Following Pulmonary and Cardiac Rehabilitation: A Repeated Measures Study

**DOI:** 10.3390/bs14100965

**Published:** 2024-10-17

**Authors:** Kristie Lee Alfrey, Benjamin Gardner, Jenni Judd, Christopher D. Askew, Corneel Vandelanotte, Amanda L. Rebar

**Affiliations:** 1Motivation of Health Behaviours Lab, Appleton Institute, Central Queensland University, Bundaberg, QLD 4670, Australia; k.alfrey@cqu.edu.au; 2Habit Application and Theory Group, Department of Psychology, University of Surrey, Guildford GU2 7XH, UK; benjamin.gardner@surrey.ac.uk; 3School of Graduate Research, Appleton Institute, Central Queensland University, Bundaberg, QLD 4670, Australia; j.judd@cqu.edu.au; 4VasoActive Research Group, School of Health, University of the Sunshine Coast, Sippy Downs, QLD 4556, Australia; caskew@usc.edu.au; 5Sunshine Coast Health Institute, Sunshine Coast Hospital and Health Service, Birtinya, QLD 4565, Australia; 6Physical Activity Research Group, Appleton Institute, Central Queensland University, Rockhampton, QLD 4702, Australia; c.vandelanotte@cqu.edu.au; 7Motivation of Health Behaviors Lab, Health Promotion, Education, and Behavior, Arnold School of Public Health, University of South Carolina, Columbia, SC 29208, USA

**Keywords:** exercise, self-determination, habit, intention, cardio-pulmonary, physical activity, motivation, rehabilitation, heart, lung

## Abstract

Background: Exercise rehabilitation programmes are important for long-term health and wellbeing among people with cardiac and pulmonary diseases. Despite this, many people struggle to maintain their physical activity once rehabilitation ends. This repeated measures study tracked changes in physical activity behaviour and motivation during and after completing a community-based exercise rehabilitation programme. Methods: Cardiac and pulmonary exercise rehabilitation patients (*N* = 31) completed six once-monthly measures of physical activity (MET·min), self-determined motivation, intention, and habit strength for rehabilitation exercise (within rehabilitation sessions) and lifestyle physical activity (outside of rehabilitation sessions). Linear regression and random effects models with estimated marginal means were used to test for associations between physical activity motivation and behaviour and change during and post-rehabilitation. Results: Overall physical activity decreased after rehabilitation (823 MET·min) despite patients becoming more self-determined for lifestyle physical activity during rehabilitation. More self-determined motivation, stronger intentions, and stronger habits were associated with more lifestyle physical activity behaviour. However, none of these motivation variables were significantly associated with rehabilitation exercise behaviour. Conclusions: Among community-based cardiac and pulmonary rehabilitation patients, physical activity levels decreased following exercise rehabilitation programmes. The findings revealed clear distinctions in the motivation of rehabilitation exercise compared to lifestyle physical activity. Exercise rehabilitation programmes might improve the longevity of outcomes by integrating approaches to enhance lifestyle physical activity beyond the clinic.

## 1. Background

Cardiac and pulmonary diseases are among the top five most prevalent and burdensome health conditions globally, accounting for more than 20% of the world’s disability and more than 3.5 million global fatalities annually [1]. Deaths from these diseases have recently declined in high-income countries due to more effective primary treatment options [2,3]. Improved survival rates mean, however, that more people than ever are living longer while managing these chronic diseases. People living with cardiac and pulmonary diseases have an increased likelihood of poor physical health, mental health, and low quality of life [4,5]. Effective management strategies are needed to ensure that people with cardiac and pulmonary disease live long, healthy, and happy lives.

Physical activity is essential for quality of life and longevity [6], particularly for people with cardiac or pulmonary diseases [7,8]. For example, regular physical activity reduces the risk of all-cause mortality by 12–14% for those with heart disease and 30% for those with chronic lung disease [9,10]. Physical activity can also offset the negative impacts of cardiac and pulmonary disease on quality of life [11,12]. Effective long-term management of cardiac and pulmonary diseases requires engagement in regular physical activity.

Following cardiac or pulmonary health events, patients are often referred to exercise rehabilitation programmes. Exercise rehabilitation programmes are typically short-term (6–12 weeks), comprised of weekly in-clinic exercise sessions designed to improve mobility, functional capacity, and fitness [13,14]. A wealth of evidence demonstrates positive health outcomes of exercise rehabilitation in cardiac and pulmonary patients, including improved physical and mental health, quality of life, and fitness [15,16,17,18]. Unsurprisingly, patients’ physical activity levels increase when enrolled in these programmes [14,19]. Some evidence suggests that people compensate for the exercise they are engaged in during rehabilitation with less lifestyle physical activity [20,21], although the findings are mixed [22,23]. If cardiac and pulmonary rehabilitation patients replace their lifestyle physical activity with rehabilitation exercises, it may put their activity levels at risk once rehabilitation programmes cease.

Understanding motivation for long-term engagement in regular physical activity is essential for increasing the longevity of the benefits of cardiac and pulmonary exercise rehabilitation. Self-determination, intention, and habit have been theorised as key predictors of physical activity, especially for the long-term maintenance of changes in physical activity behaviour [24]. Self-determination theory proposes that people are innately driven to behave in ways that lead to autonomous, or self-determined, motivation, which enhances individuals’ overall growth and wellbeing [25,26]. The quality of motivation is proposed to vary across a continuum of self-determination, from purely self-determined or intrinsic regulation (i.e., internal drives based on core values and interests) to non-self-determined, wholly extrinsic regulation (i.e., drives to behave based on external pressures). Evidence has consistently shown that people with more self-determined physical activity motivation tended to be more strongly motivated overall [27], were more likely to act on their motivation [28], and were more likely to maintain physical activity over the long term [29,30]. Previous research amongst cardiac rehabilitation patients supports the importance of self-determined motivation for long-term maintenance of physical activity [31,32].

Retaining positive physical activity intentions is also important for the motivation of regular engagement in physical activity over the long term. Intentions are proposed to represent the summation of all conscious, deliberative influences on behaviour—such as outcome expectancies, social norm beliefs, and perceptions of behavioural control [33,34]—and are positioned as the most proximal determinant of behaviour [35]. Intentions have been shown to be an important predictor of exercise and physical activity in cardiac exercise rehabilitation patients [36,37].

Habit is a core mechanism for long-term engagement in physical activity [38]. Habit can be defined as a non-conscious process through which an association between a cue (e.g., waking up) and a behavioural response (e.g., going for a walk), learned through repetition, automatically triggers impulses to act when the cue is encountered [39]. Habit is thought to sustain action even when people have little conscious motivation to act [40,41]. Evidence on the value of habit for improving physical activity in cardiac and pulmonary rehabilitation patients is promising [42,43].

Perhaps owing to the dichotomisation of long-term behaviour change into discrete ‘initiation’ and ‘maintenance’ stages [38], behaviour maintenance is tacitly conceived of as a single, stable action phase. This overlooks the potential that physical activity motivation may change over time throughout either or both the initiation and maintenance stages. For example, the extent to which a behaviour is self-motivated can change over time through praise and mastery experiences; initially, extrinsically motivated behaviours may become internalised [29], and conversely, providing extrinsic rewards for an intrinsically motivated behaviour can foster external motivation [44]. Similarly, intention strength will likely change as new barriers and facilitators of physical activity emerge [45], and the strength of newly formed positive habits can dip over the long term [46,47]. Decreases in the strength of self-determined motivation, intention, or habit are likely to precede a decline in physical activity [24]. Understanding how motivation for physical activity changes over time, both during and after exercise rehabilitation, may aid the development of interventions to support long-term physical activity maintenance among cardiac and pulmonary rehabilitation patients.

## 2. The Present Study

Exercise rehabilitation programmes are crucial aids for providing guided advice and support for safe and healthy engagement in exercise. However, little is known about changes in cardiac and pulmonary patients’ physical activity behaviour and motivation during and after exercise rehabilitation. This study was undertaken among community-based cardiac and pulmonary disease patients to investigate how rehabilitation exercise, lifestyle physical activity, self-determined motivation and habit and intention changed during and after participating in a rehabilitation exercise programme.

## 3. Methods

### 3.1. Study Design and Procedure

This repeated measures study was conducted from June to December 2018, with participants completing six once-monthly survey assessments. Participants were tracked throughout their individual journeys of the community-based rehabilitation programmes, so weeks in rehabilitation and time of completion/drop-outs were tracked for each individual. As such, the time in this study and time to and since completing rehabilitation differed for each individual.

### 3.2. Participants, Setting, and Recruitment

A non-probability convenience sampling method was utilised, in which participants were patients who were medically referred to a local community-based clinical cardiovascular and pulmonary rehabilitation programme based in a regional city in Australia (Bundaberg, Queensland) as part of usual care. Regular engagement in this particular rehabilitation programme consisted of eight weeks of one-hour clinically-based, supervised exercise sessions conducted twice a week (16 exercise sessions in total). Patients were provided with information regarding the research and invited to participate during clinical consultation. When patients met the eligibility criteria (≥18 years of age and had been referred to the rehabilitation programme) and were interested in participating in the research, a face-to-face appointment was scheduled. At this point, participants’ informed consent, preferred survey method (paper or electronic), and baseline self-reported survey data were collected. All participants who provided informed consent opted for a hard-copy (paper) survey posted to their home addresses. Following the initial face-to-face research appointment, participants were posted a paper self-report survey each month for the following four months. In month six, each patient was invited for a second face-to-face appointment and completed their final (sixth) self-report survey. After the second appointment, each participant was given an AUD 50 gift card. All procedures were approved by the associated University’s Human Research Ethics Committee (#20840).

### 3.3. Measures

In each of the six surveys, participants were asked about physical activity engaged in outside of the clinical sessions (i.e., ‘lifestyle physical activity’) and exercise engaged in during rehabilitation sessions (‘rehabilitation exercise’). Measures were scored using the R psych package 1.8.4 [48].

#### 3.3.1. Lifestyle Physical Activity

Lifestyle physical activity was measured using the International Physical Activity Questionnaire—Short Form IPAQ-SF [49]. The IPAQ-SF consists of seven items requiring participants to reflect on the past seven days and report the number of days and average time (hours and minutes) spent performing vigorous, moderate, and walking activities (e.g., “During the last 7 days, on how many days did you do vigorous physical activities like heavy lifting, digging, aerobics or fast bicycling?”; “How much time did you usually spend doing vigorous physical activities on one of those days?”). Before each item, participants were reminded not to include rehabilitation exercises. As per validated scoring procedures [49], scores were calculated for estimated metabolic equivalents (METs) per week and categories of low, moderate, or high physical activity levels.

#### 3.3.2. Rehabilitation Exercise

Rehabilitation exercises were scored using a second set of the IPAQ-SF items [49], worded to reflect the rehabilitation-based exercises and activity sessions (e.g., “During the last 7 days, on how many days did you do vigorous exercise for at least 10 min at a time, during rehabilitation sessions?”). Scoring calculations were the same as for lifestyle physical activity.

#### 3.3.3. Self-Determined Motivation

Two versions of the Behavioural Regulation in Exercise Questionnaire BREQ-2 [50] were used, one for lifestyle physical activity and one for rehabilitation exercise. The BREQ-2 uses a five-point Likert scale (0 = “Not true for me”–4 = “Very true for me”) with items such as “I exercise because it’s fun” and “I feel like a failure when I haven’t exercised”. Scores were calculated as subscales of intrinsic (four items), identified (four items), introjected (three items), and external (four items) regulation, as well as an overall relative autonomy index score. Interitem reliabilities ranged from α = 0.61 to α = 0.93.

#### 3.3.4. Intention Strength

Intentions to engage in lifestyle physical activity and rehabilitation exercise were separately measured using single items (“Over the next month, to what extent do you intend to exercise outside of rehabilitation?”, “Over the next month, to what extent do you intend on attending rehabilitation sessions?”). Response scales ranged from 1 = Not at all to 7 = A lot [51].

#### 3.3.5. Habit Strength

Habit strength was measured about lifestyle physical activity and exercise rehabilitation separately via the validated four-item Self-Report Behavioural Automaticity Index SRBAI [52], which is a subscale of the Self-Report Habit Index SRHI [53]. Response options varied on seven-point Likert-style scales (from 1 = Strongly disagree to 7 = Strongly agree) to two sets of four items, which followed the stems “Attending rehabilitation sessions is something I do…” and “Exercise is something I do…”. The items were: “…automatically”, “…without having to consciously remember”, “…without thinking”, and “…I start doing before I realise I’m doing it”. Interitem reliability was α = 0.98 for lifestyle physical activity habit strength and α = 0.90 for exercise rehabilitation habit strength.

### 3.4. Data Management and Analyses

Statistical analyses were performed in R version 3.6.2 [54]. Descriptive statistics (*M*, *SD*, intraclass correlations [ICC]) were calculated for all variables. Linear regression models were conducted to test for overall associations between maintenance determinants and behaviour for exercise rehabilitation and lifestyle physical activity during and following rehabilitation. During assumption testing, it became clear that there was the risk of undue influence from outliers of the self-reported physical activity measure, so the variable was truncated to the 75% interquartile range value for analyses. Following transformation, all model assumptions were met.

To test for differences in physical activity behaviour and motivation between rehabilitation exercise, lifestyle physical activity during rehabilitation, and lifestyle physical activity following rehabilitation, random effects models and estimated marginal means were calculated for pairwise comparisons with Tukey’s *p*-value adjustment for multiple corrections and the Kenward–Roger method for degrees of freedom [55,56]. To test for change over time during and after exercise rehabilitation, random effects models were estimated with time modelled linearly as a month in this study, separately for during and post-exercise rehabilitations. Effect sizes were estimated as marginal R-squared estimates [57], interpretable as pseudo-*R*^2^ (i.e., the proportion of variability explained by the effect). Post-hoc power calculation established that this study was powered to detect medium effects (*d* = 0.49, 1 − β = 80%, α = 0.05) at the within-person level and large effects (*d* = 0.68, 1 − β = 80%, α = 0.05) at the between-person level [58,59].

## 4. Results

### 4.1. Sample Characteristics

Thirty-one participants (cardiac patients *n* = 15; pulmonary patients *n* = 16) participated in this study. A detailed summary of participant demographic characteristics is presented in Table 1. The primary diagnosis of most patients was Chronic Obstructive Pulmonary Disease (29.0%) or Hypertension (22.6%). Most patients were male (71.0%) and Caucasian (96.8%), with an *M* age of 71.5 years (*SD* = 9.4; range = 36–84 years).

### 4.2. Change in Physical Activity Behaviour

The ICC for lifestyle physical activity behaviour was 71.9%, representing less change over time compared to between-person differences, in that about two-thirds of variability was accounted for by between-person differences. There was far more within-person variability in exercise rehabilitation physical activity (ICC of 39.1%), with only about one-third of variability accounted for by differences between people.

Descriptive statistics and the results of the estimated means difference tests are presented in Table 2. Overall, physical activity decreased after exercise rehabilitation, but this was largely due to the ceasing of the rehabilitation exercise rather than a decline in lifestyle physical activity. As shown in Table 3 and Figure 1, total physical activity behaviour (i.e., the sum of rehabilitation exercise and lifestyle physical activity) did not significantly increase during exercise rehabilitation but declined steadily once exercise rehabilitation was completed, with an estimated 3% of the variability in physical activity accounted for by the decrease across time. Lifestyle physical activity did not significantly change from during to following rehabilitation.

### 4.3. Physical Activity Behaviour and Motivation

Overall associations between physical activity motivation variables and rehabilitation exercise and lifestyle physical activity are shown in Table 4. Rehabilitation exercise was not significantly associated with any motivation variable. During the months that participants were in exercise rehabilitation, lifestyle physical activity was significantly associated with identified regulation (*Adj. R*^2^ = 11%), intention strength (*Adj. R*^2^ = 23%), and lifestyle activity habit strength (*Adj. R*^2^ = 14%). After exercise rehabilitation was complete, lifestyle physical activity was associated with intrinsic regulation (*Adj. R*^2^ = 17%), identified regulation (*Adj. R*^2^ = 36%), individuals’ relative autonomy index scores (*Adj. R*^2^ = 19%), intention strength (*Adj. R*^2^ = 45%), and lifestyle activity habit strength (*Adj. R*^2^ = 40%).

### 4.4. Self-Determined Motivation

All forms of regulation for rehabilitation exercise had ICCs between 26.6% and 43.4%, indicating that between-person differences only accounted for about one-third of the variability in regulation. So, most variability was accounted for by within-person change across this study. Comparatively, the ICCs for regulation of lifestyle physical activity were far more stable, with ICCs between 73.2% and 78.1%, indicating that most variability was due to between-person differences rather than changes over the study period.

Descriptive statistics and the results of the estimated means differences for lifestyle physical activity, rehabilitation exercise, and self-determined motivation are presented in Table 3. Intrinsic and identified regulations were stronger for rehabilitation exercise than for lifestyle physical activity both during and following rehabilitation. There were no differences in introjected or external regulation for exercise rehabilitation and lifestyle physical activity either during or following rehabilitation. There were no significant changes in any form of regulation for lifestyle physical activity from during to following rehabilitation. Overall, relative autonomy indexes were higher for rehabilitation exercise than for lifestyle physical activity both during and following rehabilitation.

When considering change over time both during and following exercise rehabilitation (Table 4), no specific form of regulation for lifestyle physical activity changed during exercise rehabilitation. However, an increase in the relative autonomy index suggested that people became more self-determined for lifestyle physical activity over time during rehabilitation. This increase over time explained 2% of the variability in the relative autonomy index. Following completion of rehabilitation, there were no significant changes in the more self-determined forms of motivation (intrinsic, introjected, and identified); however, external regulation decreased over time, with less than 1% of the variability in external regulation explained by change across time.

### 4.5. Intention

Intention strength for rehabilitation exercise had an ICC of 4.2% (95% CI = −14.2% to 28.6%), indicating very low between-person differences in intention strength (with most having quite strong intentions). Comparatively, the stability of intention strength for lifestyle physical activity was much less, with an ICC of 45.3%, indicating that about half of the variability was from between-person differences and half from within-person change. Participants reported stronger intentions for rehabilitation exercise than for lifestyle physical activity during and following rehabilitation. There were no changes in lifestyle physical activity or exercise rehabilitation intention strength from during to following rehabilitation, and there were no changes across time for rehabilitation exercise either during or following rehabilitation (Table 3 and Table 4).

### 4.6. Habit Strength

Habit strength for rehabilitation exercise had an ICC of 27.1%, reflecting considerable change across time. However, habit strength for lifestyle physical activity had an ICC of 76.5%, indicating minimal change across this study, with most variability accountable at the between-person level. Participants reported stronger habits for rehabilitation exercise than for lifestyle physical activity during or following rehabilitation. There were no significant changes in lifestyle physical activity habit strength from during to following rehabilitation, though trends indicated that habit strength tended to decrease (Table 3). There were no changes across time either during or following rehabilitation (Table 4).

## 5. Discussion

This repeated measures study tracked physical activity behaviour and motivation during and after a community-based cardiac and pulmonary exercise rehabilitation programme. Overall, physical activity levels decreased following rehabilitation as a result of the ceasing of rehabilitation exercise and no added gain in lifestyle physical activity. Uniquely, this study provided insight into the dynamics of motivation for both lifestyle physical activity (engaged in outside rehabilitation sessions) and exercise rehabilitation (performed within rehabilitation sessions), revealing that the processes underlying adherence to exercise rehabilitation differed from those for lifestyle physical activity. Unless more is achieved to enhance motivation for lifestyle physical activity post-rehabilitation, the benefits of exercise rehabilitation will be short-lived.

Patients’ lifestyle physical activities changed very little across this study, demonstrating that engaging in rehabilitation exercises did not replace their other physical activity behaviours. This finding adds to the evidence that not all exercise rehabilitation sessions lead to a compensatory effect of less physical activity outside of rehabilitation [22,23]. Given the conflicting evidence on this phenomenon, more is needed to understand what factors influence whether a person does or does not compensate for rehabilitation exercise with less lifestyle physical activity. Some evidence suggests that compensation may be more likely with older populations or when mixed aerobic and resistance training is used, but even these findings are not reliable across all studies [22]. When patients enrol in exercise rehabilitation programmes, care is needed to encourage rehabilitation exercise and lifestyle physical activity throughout programme participation.

Support for lifestyle physical activity becomes even more essential in the transition that follows the completion of exercise rehabilitation programmes. In our study, when including rehabilitation exercise, patients’ overall physical activity levels increased during programme enrolment, echoing evidence showing an increase in physical activity during such programmes [14,19]. Notably, however, patients did not increase their lifestyle physical activity behaviours once their rehabilitation exercise programmes ended. The substantial health and wellbeing outcomes gained from participating in exercise rehabilitation, e.g., [15,17], are unlikely to continue beyond the time of the rehabilitation programme unless changes are made to enhance motivation for lifestyle physical activity amidst and following the rehabilitation programme. These study findings highlight the need for a focus on motivation intervention and boosters within exercise rehabilitation programmes, which are in line with contemporary guidelines which advocate for self-management rather than a reliance on clinical management of chronic diseases [60].

One of the novel elements of this study was the separate investigation of motivation for rehabilitation exercise vs. lifestyle physical activity, and the results show large differences in these motivational processes. Overall, patients’ motivations for lifestyle physical activity and rehabilitation exercise were generally high, with self-determination, intention strength, and habit strength for exercise rehabilitation rated consistently higher than for lifestyle physical activity. The fact that patients were so strongly motivated for exercise rehabilitation was encouraging, given the evidenced benefits of such programmes [14,19]. However, this motivation did not necessarily translate into more physical activity.

For lifestyle physical activity, more self-determined motivation, stronger intentions, and stronger habits were associated with greater behavioural engagement. This aligns with theory and evidence that these aspects of motivation are important in influencing physical activity behaviour, e.g., [61,62,63,64]. However, in contrast to theory and evidence, none of these motivation variables were associated with rehabilitation exercise behaviour. The lack of motivation–behaviour associations found for rehabilitation exercises could be the byproduct of a lack of variability in motivation, resulting in a ceiling effect or measurement artefact. Energy expenditure was self-reported based on estimated time spent in different intensities of activity during sessions, which is likely prone to error. It may be that motivation would be associated with less subjective measures of exercise energy expenditure, such as accelerometer monitors or physiological trackers. Future research is needed to test the replicability of our findings for clarity on the motivational processes of rehabilitation exercise.

If our findings of rehabilitation exercise behaviour not being associated with intentions, habits, or self-determined motivation for rehabilitation exercises are found to be reproducible, it has important implications for clinical practice. The aspects of motivation that influence lifestyle physical activity may be distinct from those for exercise behaviour within rehabilitation programmes. Most research has considered motivation for adherence to rehabilitation exercise [65] or motivation for lifestyle physical activity following rehabilitation, e.g., [66]. More research is needed to track motivation for these two different types of activity as people transition into the post-rehabilitation phase.

Our findings add to the limited understanding of what happens to physical activity motivation across the transition from having regular exercise rehabilitation to the programme ceasing. We found that there was very little change in people’s motivations for lifestyle physical activity either during or following patients’ involvement in exercise rehabilitation. This finding aligns with other evidence of a lack of change in motivation for physical activity across rehabilitation programmes [67]. It seems that patients maintain their levels of self-determination, intention, and habit strength for physical activity during and beyond the completion of exercise rehabilitation. For those who have strong habits, intentions, and self-determination for lifestyle physical activity, this finding is encouraging in that their motivation is unlikely to decline due to rehabilitation. However, for patients with low motivation, it may be a missed opportunity to improve people’s motivation for lifestyle physical activity [68].

Most exercise rehabilitation programmes focus on in-gym exercise training with less focus on motivation and behaviour change strategies [69]. If exercise rehabilitation were to include motivational training and support for safely enhancing physical activity levels beyond exercise rehabilitation, the physical and mental health benefits of exercise rehabilitation could have long-lasting benefits for patients at risk of inactivity following exercise rehabilitation. Simple planning aides have been shown to be effective in enhancing physical activity habit strength and in turning strong intentions into physical activity behaviour [70,71]. For example, one study found that an exercise rehabilitation programme with planning intervention elements that helped people specify when, where, and how they would engage in physical activity (action plans), as well as how to deal with anticipated barriers (coping plans), led to increases in physical activity 2 months following discharge [72]. When applied in the incremental distribution with telephone-based boosters, planning aides can enhance motivation for physical activity and physical activity habit strength for more than a year following exercise rehabilitation programmes [73]. Further exploration is needed to determine how to effectively implement these and other potentially effective approaches for utilising the time clinicians have with exercise rehabilitation patients to enhance patients’ physical activity motivations.

### 5.1. Conclusions

Physical activity is a cost-effective way to enhance cardiac and pulmonary patients’ quality and longevity of life [7,8], and exercise rehabilitation plays an important role in helping people engage in physical activity that is safe and beneficial. Our findings show that once rehabilitation programmes end, patients’ physical activity levels decline, and their motivation for lifestyle physical activity does not improve. To ensure the longevity of the benefits of exercise rehabilitation for this important and widely prevalent at-risk population, more work is needed to enhance lifestyle physical activity motivation and behaviour amidst and following cardiac and pulmonary rehabilitation programmes.

### 5.2. Study Limitations and Biases

As a result of the intensive, rich person-level focus of this study, the data were from a small sample of relatively demographically homogenous patients from one exercise rehabilitation clinic. The non-probability convenience sampling method limits generalisability; the findings of this study should not be generalised to the target population without further research with a probability sample. Geographical location, socio-demographic factors, and clinic-specific variables may have systematically impacted results. Physical activity was self-reported, which may elicit some measurement artefact from response bias.

### 5.3. Future Directions

Future research efforts are needed to test the replicability and generalisability of our findings, with consideration of the limitations of our study. Future research is needed to test replicability using data from different locations, communities, and clinics. Most of our study sample (86%) was older than 60 years, so future research should consider whether these findings generally work for younger populations as well. Our study focused on patients with any cardiac and pulmonary condition. Future research might tease out whether there are different patterns of physical activity behaviour or motivation between patients living with different cardiac or pulmonary diseases. Consideration is also needed to ensure such research is person-focused rather than condition-focused, given that physical and mental health conditions are often comorbid [74].

## Figures and Tables

**Figure 1 behavsci-14-00965-f001:**
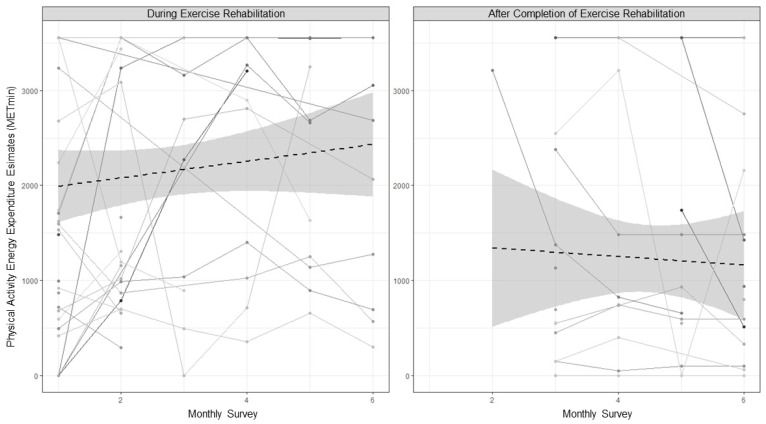
Monthly linear change in physical activity during and following exercise rehabilitation for individuals (grey lines and dots) and sample mean trends (dashed black line and shaded confidence intervals).

**Table 1 behavsci-14-00965-t001:** Summary of participant demographic characteristics.

		*n* (*N* = 31)	%
Rehabilitation Group	Pulmonary	16	51.6
	Cardiac	15	48.4
Primary Diagnosis	Chronic Obstructive Pulmonary	9	29.0
	Disease		
	Coronary Artery Bypass Graft	1	3.2
	Diabetes	1	3.2
	Emphysema	4	12.9
	High Cholesterol	2	6.5
	Hypertension	7	22.6
	Obstructive Sleep Apnea	3	9.7
	Data not Available	4	12.9
Gender	Male	22	71.0
	Female	9	29.0
Age (years)	36–48	1	3.0
	49–60	1	3.0
	61–72	14	45.0
	73–84	15	48.0
Marital Status	Married	20	64.5
	Partnered	4	12.9
	Single	7	22.6
Employment Status	Retired	26	83.9
	Unemployed	2	6.5
	Other	3	9.7
Highest Level of Education	High School (≤Year 12)	20	64.5
	TAFE/Certificate/Diploma	2	6.5
	University Degree	5	16.1
	Post Graduate Degree	4	12.9
Ethnicity	White	30	96.8
	Other	1	3.2

**Table 2 behavsci-14-00965-t002:** Physical activity behaviour and maintenance determinants for rehabilitation exercise and lifestyle physical activity out of rehabilitation during and following rehabilitation.

Variable	Rehabilitation Exercise	For Lifestyle Physical Activity (Outside of Rehabilitation)	Δ	
		During Rehabilitation	Following Rehabilitation	Significant Differences
	*M* (*SD*)	*M* (*SD*)	*M* (*SD*)	
Self-reported physical activity behaviour	604.12 (495.54)	1165.74 (683.43)	946.79 (710.31)	Rehab Ex < PA Dur Rehab Ex < PA Post PA Dur = PA Post
Intrinsic regulation	3.35 (0.87)	2.46 (1.25)	2.34 (1.15)	Rehab Ex < PA Dur Rehab Ex < PA Post PA Dur = PA Post
Introjected regulation	0.80 (1.05)	1.11 (1.15)	0.78 (1.11)	Rehab Ex = PA Dur Rehab Ex = PA Post PA Dur = PA Post
Identified regulation	3.22 (0.55)	2.99 (0.90)	2.57 (1.04)	Rehab Ex < PA Dur Rehab Ex < PA Post PA Dur = PA Post
External regulation	0.31 (0.70)	0.69 (1.03)	0.22 (0.58)	Rehab Ex = PA Dur Rehab Ex = PA Post PA Dur = PA Post
Relative autonomy	14.82 (4.60)	9.86 (6.82)	9.85 (6.47)	Rehab Ex < PA Dur Rehab Ex < PA Post PA Dur = PA Post
Intention strength	5.99 (1.74)	4.57 (1.89)	4.35 (1.91)	Rehab Ex < PA Dur Rehab Ex < PA Post PA Dur = PA Post
Habit strength	5.73 (1.19)	3.98 (1.93)	3.46 (1.85)	Rehab Ex < PA Dur Rehab Ex < PA Post PA Dur = PA Post

Note. Physical activity reported in estimated energy expenditure per week. Rehab Ex = Rehabilitation exercise; PA Dur = Lifestyle physical activity during rehabilitation; PA Post = Lifestyle physical activity following rehabilitation.

**Table 3 behavsci-14-00965-t003:** Linear Change in Physical Activity Behaviour and Maintenance Determinants for Lifestyle Physical Activity Out of Rehabilitation During and Following Rehabilitation.

Outcome	Monthly Change During Exercise Rehabilitation	Monthly Change After Exercise Rehabilitation
	Estimate (95% CI)	Estimate (95% CI)
Physical activity behaviour	67.83 (−42.11 to 179.42)	−199.74 (−362.12 to −27.37) *
Intrinsic regulation	0.04 (−0.03 to 0.12)	−0.07 (−0.18 to 0.04)
Introjected regulation	−0.03 (−0.11 to 0.05)	−0.05 (−0.17 to 0.05)
Identified regulation	0.06 (−0.00 to 0.12)	−0.07 (−0.20 to 0.06)
External regulation	−0.05 (−0.13 to 0.03)	−0.03 (−0.06 to −0.00) *
Relative autonomy	0.65 (0.23 to 1.05) *	0.14 (−0.50 to 0.79)
Intention strength	−0.10 (−0.30 to 0.10)	0.32 (−0.00 to 0.66)
Habit strength	0.03 (−0.10 to 0.16)	0.17 (−0.03 to 0.38)

Note. Physical activity reported in estimated energy expenditure per week, including both rehabilitation exercise and lifestyle physical activity. All motivation variables are for lifestyle physical activity only (excluding exercise rehabilitation). * *p* < 0.05.

**Table 4 behavsci-14-00965-t004:** Associations Between Physical Activity Behaviour and Maintenance Determinants for Rehabilitation Exercise and Lifestyle Physical Activity Out of Rehabilitation During and Following Rehabilitation.

Predictor	Rehabilitation Exercise	Lifestyle Physical Activity During Rehabilitation	Lifestyle Physical Activity After Completion of Rehabilitation
	Estimate (95% CI)	Estimate (95% CI)	Estimate (95% CI)
Intrinsic Regulation	−109.60 (−379.06 to 159.77)	87.05 (−67.94 to 242.04)	237.80 * (15.88 to 459.64)
Introjected Regulation	201.90 (−12.61 to 416.42)	157.43 (−15.44 to 330.21)	191.00 (−35.53 to 417.59)
Identified Regulation	97.47 (−346.28 to 541.22)	213.92 * (15.30 to 412.53)	378.00 * (151.86 to 604.23)
External Regulation	45.33 (−183.90 to 274.56)	209.60 (−16.53 to 435.83)	−0.62 (−330.96 to 329.71)
Relative Autonomy Index	−24.65 (−77.51 to 28.22)	14.07 (−14.19 to 42.33)	43.14 * (5.47 to 80.81)
Intention Strength	−39.41 (−187.97 to 109.16)	157.18 * (52.77 to 261.60)	216.72 * (104.08 to 329.36)
Habit Strength	−57.80 (−257.49 to 141.90)	118.64 * (20.14 to 217.15)	205.00 * (92.42 to 317.57)

Note. Outcome is physical activity reported in estimated energy expenditure per week. * *p* < 0.05.

## Data Availability

The data presented in this study are available upon request from the corresponding author due to restrictions from the ethics committee to reduce the risk of participants’ loss of confidentiality.

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
