# Peer review of "Physical Activity Behaviour and Motivation During and Following Pulmonary and Cardiac Rehabilitation: A Repeated Measures Study"

_behavsci, 2024, doi:10.3390/bs14100965_

Round 1

Reviewer 1 Report

Comments and Suggestions for Authors

Thank you for the opportunity to review this paper.

The aim of this study is to evaluate the changes of lifestyle physical activity, motivation and intention during and after exercise rehabilitation programs in 31 patients with history of cardiac or pulmonary disease. The results of this are important for future studies with more patients, from more centers, and with an impact on the exercise rehabilitation programs. 

Minor comments:

- The authors have mentioned that the patients of this study had a "heart or lung disease diagnostic or surgery "  (line 135). Please reformulate and , if possible, add more data about patiets history, about the need and the duration of hospitalization, the coservative or interventional approach (surgery or minimal invasive techniques i.e diadermic intervention after an acute myocardial infarction), or comorbidities with impact on the rehabilitation program and the behavioural changes.

- 96% of the patients of this study were >60 years old. Please add a comment about the changes observed in the younger patients, correlated with their demographic data and specify future directions for this category

- Please review the sructure of this paper (the numbers of paragraphs beginning with the third (3. Methods, 3.1.1 etc) and reformulate in a separate paragraph the limitations of this study

Author Response

Thank you for the opportunity to review this paper.

The aim of this study is to evaluate the changes of lifestyle physical activity, motivation and intention during and after exercise rehabilitation programs in 31 patients with history of cardiac or pulmonary disease. The results of this are important for future studies with more patients, from more centers, and with an impact on the exercise rehabilitation programs. 

Authors’ response: We appreciate the time and effort taken by the reviewer.

Minor comments:

- The authors have mentioned that the patients of this study had a "heart or lung disease diagnostic or surgery "  (line 135). Please reformulate and , if possible, add more data about patiets history, about the need and the duration of hospitalization, the coservative or interventional approach (surgery or minimal invasive techniques i.e diadermic intervention after an acute myocardial infarction), or comorbidities with impact on the rehabilitation program and the behavioural changes.

Authors’ response: We do not have data on the treatment received by patients; however, we do have information on their primary diagnosis, which we have added to the revised manuscript (lines 224-225 and Table 1). We also revised the statement referred to in the reviewer’s comment to clarify that: “Patients were medically referred to a local community-based clinical cardiovascular and pulmonary rehabilitation program based in a regional city in Australia (Bundaberg, Queensland), as part of usual care. (lines 135-136)”

- 96% of the patients of this study were >60 years old. Please add a comment about the changes observed in the younger patients, correlated with their demographic data and specify future directions for this category

Authors’ response: The reviewer makes a great point about the lack of younger patients being represented in this study. Although, our data would not allow for inferential statistics based on age, we included a statement suggesting future research consider this: “Most of our study sample (86%) was older than 60 years, so future research should consider whether these findings generalize to younger populations as well. (lines 435-436)”

- Please review the sructure of this paper (the numbers of paragraphs beginning with the third (3. Methods, 3.1.1 etc) and reformulate in a separate paragraph the limitations of this study

Authors’ response: We have revised the numbering of the paragraphs and adjusted structure as recommended, with a new Study Limitations section featured as 5.2. (lines 424-428).

Reviewer 2 Report

Comments and Suggestions for Authors

I have received for review an article entitled ,, Physical Activity Behaviour and Motivation During and Fol-2 lowing Pulmonary and Cardiac Rehabilitation: A Repeated 3 Measures Study” which is being processed by Behavioral Sciences.

The proposed manuscript is one with therapeutic and prognostic impact, but I encourage authors to address the following issues to improve its quality:

Abstract -summarizes the main elements analyzed in the manuscript, emphasizing the prognostic implications of cardiac rehabilitation. I encourage authors to specify more keywords in order to increase the dissemination rate of the manuscript.

Introduction - mentions the main existing data in the specialized literature related to the topic of the manuscript. Epidemiologic data were provided, but it would be useful in relation to the negative clinical and the negative implications of sedentary lifestyle and failure to refer patients with CV and pulmonary pathologies to specialized centers. The purpose of the article should also be integrated in this section.

Section 2

Line 135 - mention the pathologies for which the rehabilitation program was recommended

The period of the study and the total number of patients who underwent rehabilitation programs and those who agreed to participate in the study should be mentioned.

Programs used for statistical processing and ethical issues should be mentioned in 2 separate sections.

The results section has been well produced and presents the main issues identified

Discussion - presents a comparative analysis of the data obtained with similar data from the literature

 Pay attention to the formatting of the manuscript

In conclusion, the proposed manuscript brings to attention an extremely interesting topic, presenting scientific information with therapeutic and prognostic value, but needs revision in order to be considered for publication.

Author Response

I have received for review an article entitled ,, Physical Activity Behaviour and Motivation During and Fol-2 lowing Pulmonary and Cardiac Rehabilitation: A Repeated 3 Measures Study” which is being processed by Behavioral Sciences.

The proposed manuscript is one with therapeutic and prognostic impact, but I encourage authors to address the following issues to improve its quality:

Authors’ response: Thank you for the peer review and recommendations.

Abstract -summarizes the main elements analyzed in the manuscript, emphasizing the prognostic implications of cardiac rehabilitation. I encourage authors to specify more keywords in order to increase the dissemination rate of the manuscript.

              Authors’ response: Thank you. We have added more keywords:

Keywords: exercise; self-determination; habit; intention; cardio-pulmonary; physical activity; motivation; rehabilitation; heart, lung (lines 42-44)”

Introduction - mentions the main existing data in the specialized literature related to the topic of the manuscript. Epidemiologic data were provided, but it would be useful in relation to the negative clinical and the negative implications of sedentary lifestyle and failure to refer patients with CV and pulmonary pathologies to specialized centers. The purpose of the article should also be integrated in this section.

Authors’ response: We elected not to action a change to the manuscript in line with this response. The introduction already specifically describes the risks associated with inactivity (lines 56-61). We feel that adding further to the point about the systematic failure to refer patients to specialized centers would distract readers from the focus of our study, which was on individual motivation and behavior change. Additionally, we have already specified the purpose of the study in ‘The Present Study’ section (lines 120-127).

Section 2

Line 135 - mention the pathologies for which the rehabilitation program was recommended

Authors’ response: Thank you for the recommendation. we have added information on the patients’ primary diagnosis to the revised manuscript (lines 224-225 and Table 1). We also revised the statement referred to in the reviewer’s comment to clarify that: “Patients were medically referred to a local community-based clinical cardiovascular and pulmonary rehabilitation program based in a regional city in Australia (Bundaberg, Queensland), as part of usual care. (lines 135-136)”

The period of the study and the total number of patients who underwent rehabilitation programs and those who agreed to participate in the study should be mentioned.

We have added the period of data collection but do not have access to the number of patients who underwent rehabilitation programs but did not agree to participate in the study so could not include that information: “This repeated measures study was conducted from June to December 2018, with participants completing six once-monthly survey assessments. (lines 137-138)”

Programs used for statistical processing and ethical issues should be mentioned in 2 separate sections.

We have not actioned any changes to this comment because we have described the statistical processing program used separately from the ethical issues already. Software is described on line 201 in section 3.4. Ethical issues are described in lines section 3.2 on lines 151-154.

The results section has been well produced and presents the main issues identified

Authors’ response: Thank you.

Discussion - presents a comparative analysis of the data obtained with similar data from the literature

Authors’ response: Thank you.

 Pay attention to the formatting of the manuscript

Authors’ response: We have revised the formatting of the manuscript. Specifically, we included paragraph indices and revised the numbering of the section sub-headings.

In conclusion, the proposed manuscript brings to attention an extremely interesting topic, presenting scientific information with therapeutic and prognostic value, but needs revision in order to be considered for publication.

              Authors’ response: Thank you.

Round 2

Reviewer 2 Report

Comments and Suggestions for Authors

The proposed manuscript has been improved and can be considered for publication.

Author Response

Thank you for the peer review.